# $\rho$-POMDPs have Lipschitz-Continuous $\epsilon$-Optimal Value Functions

**Mathieu Fehr**[1], **Olivier Buffet**[2], **Vincent Thomas**[2], **Jilles Dibangoye**[3]

[1] École Normale Supérieure de la rue d'Ulm, Paris, France

[2] Université de Lorraine, CNRS, Inria, LORIA, Nancy, France

[3] Université de Lyon, INSA Lyon, Inria, CITI, Lyon, France

mathieu.fehr@ens.fr, olivier.buffet@loria.fr, vincent.thomas@loria.fr, jilles.dibangoye@inria.fr

## Abstract

Many state-of-the-art algorithms for solving Partially Observable Markov Decision Processes (POMDPs) rely on turning the problem into a "fully observable" problem—a belief MDP—and exploiting the piece-wise linearity and convexity (PWLC) of the optimal value function in this new state space (the belief simplex $\Delta$). This approach has been extended to solving $\rho$-POMDPs—*i.e.*, for information-oriented criteria—when the reward $\rho$ is convex in $\Delta$. General $\rho$-POMDPs can also be turned into "fully observable" problems, but with no means to exploit the PWLC property. In this paper, we focus on POMDPs and $\rho$-POMDPs with $\lambda_\rho$-Lipschitz reward function, and demonstrate that, for finite horizons, the optimal value function is Lipschitz-continuous. Then, value function approximators are proposed for both upper- and lower-bounding the optimal value function, which are shown to provide uniformly improvable bounds. This allows proposing two algorithms derived from HSVI which are empirically evaluated on various benchmark problems.

## 1 Introduction

Many state-of-the-art algorithms for solving Partially Observable Markov Decision Processes (POMDPs) rely on turning the problem into a "fully observable" problem—namely a belief MDP—and exploiting the piece-wise linearity and convexity (PWLC) of the optimal value function [Sondik, 1971, Smallwood and Sondik, 1973] in this new problem's state space (here the belief space $\Delta$). State of the art off-line algorithms [Pineau et al., 2006, Smith and Simmons, 2004] maintain approximators that (i) are upper or lower bounds, and (ii) have generalization capabilities: a local update at $b$ improves the bound in a surrounding region of $b$. This approach has been extended to solving $\rho$-POMDPs as belief MDPs—*i.e.*, problems whose performance criterion depends on the belief (*e.g.*, active information gathering)—when the reward $\rho$ is convex in $\Delta$ [Araya-López et al., 2010].[1] Yet, it does not extend to problems with non-convex $\rho$—*e.g.*, (i) if a museum monitoring system is rewarded for each visitor located with "enough certainty" (*i.e.*, using a threshold function), or (ii) if collecting data regarding patients while preserving their privacy by discarding information that could harm anonymity.

Generalizing value function approximators are also an important topic in (fully observable, mono-agent) reinforcement learning, as recently with Deep RL [Mnih et al., 2013]. To allow for error-bounded approximations in continuous settings, some works have built on the hypothesis that the dynamics and the reward function were Lipschitz-continuous (LC), which leads to Lipschitz-continuous value functions [Laraki and Sudderth, 2004, Hinderer, 2005, Fonteneau et al., 2009,

Rachelson and Lagoudakis, 2010, Dufour and Prieto-Rumeau, 2012]. Ieong et al. [2007] also considered exploiting the LC property in heuristic search settings. These approaches cannot be applied to the aforementioned partially observable problems as the dynamics of the induced MDPs are *a priori* not LC.

This paper shows that, for $\rho$-POMDPs with $\lambda_\rho$-Lipschitz reward function (and thus for any POMDP) and for finite horizons, the optimal value function is still LC, a property that shall replace the PWLC property. Yet, to allow for better approximators and tighter theoretical bounds, we use an extended definition of Lipschitz-continuity where (i) the Lipschitz constant is a vector rather than a scalar, and (ii) we consider local—rather than uniform—LC. From there, value function approximators are proposed for both upper- and lower-bounding the optimal value function. Following Smith [2007], these approximators are shown to provide uniformly improvable bounds [Zhang and Zhang, 2001] for use with point-based algorithms like HSVI, which is then guaranteed to converge to an $\epsilon$-optimal solution. This allows proposing two algorithms derived from HSVI: (i) one that uses guaranteed/safe Lipschitz constants, but at the cost of overly pessimistic error bounds, and (ii) one that searches for good Lipschitz constants, but then losing optimality guarantees. This work is also a step towards solving partially observable stochastic games as continuous-space SGs with LC approximators.

The paper is organized as follows. Section 2 discusses related work on inforation-oriented control. Sec. 3 presents background on POMDPs, $\rho$-POMDPs and Lipschitz continuity. Sec. 4 demonstrates that, for finite horizons, the optimal value function is Lipschitz-continuous, Sec. 5 proposes value function approximators and two point-based algorithms (based on HSVI). Sec. 6 evaluates them empirically. Proofs are provided as supplementary material.

## 2  Related Work

Early research on information-oriented control (IOC) involved problems formalized either (i) as POMDPs (as Egorov et al. [2016] did recently, since an observation-dependent reward can be trivially recast as a state-dependent reward), or (ii) with belief-dependent rewards (and mostly ad-hoc solution techniques). $\rho$-POMDPs allow easily formalizing many—if not most—IOC problems. Araya-López et al. [2010] show that a $\rho$-POMDP with convex belief-dependent reward $\rho$ can be solved with modified point-based POMDP solvers exploiting the PWLC property (with error bounds that depend on the quality of the PWLC-approximation of $\rho$).

The POMDP-IR framework [Spaan et al., 2015] allows describing IOC problems with linear rewards—thus, a subclass of "PWLC" $\rho$-POMDPs (i.e., when $\rho$ is PWLC). Yet, as Satsangi et al. [2015] showed that a PWLC $\rho$-POMDP can be turned into a POMDP-IR, both classes are in fact equivalent. In both cases the proposed solution techniques are modified POMDP solvers, and it seems (to us) that an algorithm proposed in one framework should apply with limited changes in the other framework. For its part, the general $\rho$-POMDP framework allows formalizing more problems—e.g., directly specifying an entropy-based criterion. While Spaan et al. [2015] obtain better empirical results with their POMDP-IR-based method than with a $\rho$-POMDP-based method, this probably says more about particular solutions applied on a particular problem than about the frameworks themselves (as discussed above).

The case of non-convex $\rho$ (including information-averse scenarios) may have been mostly avoided up to now because no satisfying solution technique existed. The present work analyzes the optimal value function's properties when $\rho$ is Lipschitz-continuous, which leads to a prototype solution algorithm. This is a first step towards proposing new tools for solving a wider class of information-oriented POMDPs than currently feasible. Future work will thus be more oriented towards practical applications, possibly with evaluations on surveillance problems—which are only motivating scenarios in the present paper. Note that Egorov et al. [2016] propose solutions dedicated to surveillance (with an adversarial setting), not for general IOC problems. Regarding adversarial settings, another very promising direction is exploiting the Lipschitz continuity in a similar manner to solve (zero-sum) Partially Observable Stochastic Games.

# 3 Background

**Notations:** We denote: $\hat{\boldsymbol{x}} = \boldsymbol{x}/\|\boldsymbol{x}\|_1$ the normalization of a vector $\boldsymbol{x}$; $|\boldsymbol{x}|$ a component-by-component (CbC) absolute value operator; $\vec{\max}_{\boldsymbol{x}} f(\boldsymbol{x})$ a CbC maximum operator for vector-valued function $f(\boldsymbol{x})$; and $\mathbf{1}$ a row vector of 1s.

## 3.1 POMDPs

A POMDP [Astrom, 1965] is defined by a tuple $\langle \mathcal{S}, \mathcal{A}, \mathcal{Z}, P, r, \gamma, b_0 \rangle$, where $\mathcal{S}$, $\mathcal{A}$ and $\mathcal{Z}$ are finite sets of states, actions and observations; $P_{a,z}(s, s')$ gives the probability of transiting to state $s'$ and observing observation $z$ when applying action $a$ in state $s$ ($P_{a,z}$ is an $\mathcal{S} \times \mathcal{S}$ matrix); $r(s,a) \in \mathbb{R}$ is the reward associated to performing action $a$ in state $s$; $\gamma \in [0; 1)$ is a discount factor; and $b_0$ is the initial belief state—*i.e.*, the initial probability distribution over possible states. The objective is then to find a policy $\pi$ that prescribes actions depending on past actions and observations so as to maximize the expected discounted sum of rewards (here with an infinite temporal horizon).

To that end, a POMDP is often turned into a belief MDP $\langle \Delta, \mathcal{A}, T, r, \gamma, b_0 \rangle$ where $\Delta$ is the simplex of possible belief states, $\mathcal{A}$ is the same action set, and $T(b, a, b') = P(b'|b, a)$ and $r(b, a) = \sum_s b(s) r(s, a)$ are the induced transition and reward functions. This setting allows considering policies $\pi \colon \Delta \to \mathcal{A}$, each being associated to its value function $V^\pi(b) \doteq E[\sum_{t=0}^\infty \gamma^t r(b_t, \pi(b_t)) | b_0 = b]$. Optimal policies maximize $V^\pi$ in all belief states reachable from $b_0$. Their value function $V^*$ is the fixed point of Bellman's *optimality* operator ($\mathcal{H}$) [Bellman, 1957] $\mathcal{H}V \colon b \mapsto \max_a [r(b, a) + \gamma \sum_z \|P_{a,z} b\|_1 V(b^{a,z})]$, and acting greedily with respect to $V^*$ provides such a policy.

$V^*$ being piece-wise linear and convex (PWLC)[2] for any finite horizon [Sondik, 1971, Smallwood and Sondik, 1973] allows to approximate it from below by an upper-envelope $U$ of hyperplanes, and from above by a lower-envelope $L$ of points. A local update at belief state $b$ then allows improving $U$ or $L$ not only at $b$ but in its vicinity. This generalization allows for error-bounded approximations using a finite number of belief points [Pineau et al., 2006], and for more efficient branch pruning in heuristic search approaches [Smith, 2007]. All this led to current off-line *point-based* algorithms such as PBVI [Pineau et al., 2003, 2006], HSVI [Smith and Simmons, 2004, 2005, Smith, 2007], SARSOP [Kurniawati et al., 2008], GapMin [Poupart et al., 2011], and PGVI [Zhang et al., 2014]. We shall consider in particular HSVI (*Heuristic Search Value Iteration*, see Algorithm 1) as it is a prototypical algorithm maintaining both $U$ and $L$, and providing performance guarantees by stopping when $U(b_0) - L(b_0)$ is below an $\epsilon$ threshold. HSVI decides on where to perform updates by generating trajectories picking (i) actions greedily w.r.t. to $U$ and (ii) observations so as to reduce the gap between $U$ and $L$. Importantly, the HSVI framework is based on uniformly improvable bounds (*cf.* Sec. 5.1) and applicable beyond POMDPs with PWLC approximations.

## 3.2 $\rho$-POMDPs

$\rho$-POMDPs [Araya-López et al., 2010] differ from POMDPs in their reward function $\rho(b, a)$—rather than $r(s, a)$—that allows defining not only control-oriented criteria, but also information-oriented ones, thus generalizing POMDPs. Such problems are met regularly, but often modeled and addressed with ad-hoc techniques [Fox et al., 1998, Mihaylova et al., 2006]. Araya-López et al. [2010] have shown that, (i) if $\rho$ is PWLC, previously described techniques can still be applied with similar error bounds, and (ii) if $\rho$ is convex and either Lipschitz-continuous or $\alpha$-Hölder (as Shannon's entropy), then a PWLC approximation of $\rho$ can be used to obtain error-bounded solutions again.

While many problems can be modeled with convex $\rho$, this leaves us with a number of problems that cannot be solved with similar approximations. Here, we will exploit Lipschitz-continuous reward functions $\rho$ to solve more general $\rho$-POMDPs with similar algorithmic schemes. As an example, in the museum monitoring scenario, with $X$ the random variable for a visitor's location and $b_X$ the corresponding belief, then $\rho_X(b, a) \doteq \sigma(\alpha(\|b_X\|_\infty - \beta))$—with $\sigma(\cdot)$ the sigmoid function—is a smooth threshold function (thus non-convex) whose Lipschitz constant depends on $\alpha > 0$ and rewarding preferably distributions whose maximum probability is greater than $\beta \in [0, 1]$.

**Algorithm 1:** Heuristic Search Value Iteration   &amp;   Inc-lc-HSVI

| | |
|---|---|
| 1  **Fct** HSVI $(\epsilon)$ | 14  **Fct** Update $(b)$ |
| 2      Initialize $L$ and $U$ | 15      $L \leftarrow$ Update $(L, b)$ |
| 3      **while** $(U(b_0) - L(b_0)) > \epsilon$ **do** | 16      $U \leftarrow$ Update $(U, b)$ |
| 4         RecursivelyTry $(b_0, d = 0)$ | |
| 5      **return** $L$ | /\* Note: Vanilla HSVI for POMDPs uses PWLC approximators. lc-HSVI is HSVI with LC approximators. \*/ |
| 6  **Fct** RecursivelyTry $(b, d)$ | |
| 7      **if** $(U(b) - L(b)) > \gamma^{-d}\epsilon$ **then** | |
| 8         Update $(b)$ | /\* Below: Main loop of Incremental lc-HSVI (see Sec. 5.2). \*/ |
| 9         $a^* \in \arg\max_{a \in \mathcal{A}}\{r(b,a) + \gamma \sum_z \|P_{a,z}b\|_1 U(b^{a,z})\}$ | 17  **Fct** inc-lc-HSVI $(\epsilon, \lambda_0)$ |
| 10        $z^* \in \arg\max_{z \in \mathcal{Z}}\{\|P_{a^*,z}b\|_1 \times (U(b^{a^*,z}) - L(b^{a^*,z}) - \gamma^{-d}\epsilon)\}$ | 18      $\lambda \leftarrow \lambda_0$ |
| 11         RecursivelyTry $(b^{a^*,z^*}, d+1)$ | 19      **while** *fails*(lc-HSVI $(\epsilon, \lambda)$) **do** |
| 12         Update $(b)$ | 20         $\lambda \leftarrow 2\lambda$ |
| 13      **return** | 21      **return** $L$ |

### 3.3 Lipschitz-Continuities (in normed spaces)

Let $f : X \to Y$ be a function, where $X$ and $Y$ are normed spaces. $f$ is *uniformly* Lipschitz-continuous if there exists $\lambda_f \in \mathbb{R}^+$ such that, for all $(\boldsymbol{x}, \boldsymbol{x}') \in X^2$, $\|f(\boldsymbol{x}) - f(\boldsymbol{x}')\| \leq \lambda_f \|\boldsymbol{x} - \boldsymbol{x}'\|$. $f$ is *locally* Lipschitz-continuous if, for each $\boldsymbol{x}$, there exists $\lambda_f(\boldsymbol{x}) \in \mathbb{R}^+$ such that, for all $\boldsymbol{x}' \in X$, $\|f(\boldsymbol{x}) - f(\boldsymbol{x}')\| \leq \lambda_f(\boldsymbol{x})\|\boldsymbol{x} - \boldsymbol{x}'\|$. The former definition is more common and induces uniform continuity of $f$, but we will rely on the later (omitting "locally"), which induces local continuity of $f$, to handle more problems and obtain tighter bounds. We propose another generalization using vector rather than scalar Lipschitz constants, again to allow for tighter bounds: $f$ is Lipschitz-continuous if, for each $\boldsymbol{x}$, there exists a *row* vector $\boldsymbol{\lambda}_f(\boldsymbol{x}) \in (\mathbb{R}^+)^{dim(X)}$ such that, for all $\boldsymbol{x}' \in X$, $\|f(\boldsymbol{x}) - f(\boldsymbol{x}')\| \leq \boldsymbol{\lambda}_f(\boldsymbol{x}) \cdot |\boldsymbol{x} - \boldsymbol{x}'|$ (scalar product equivalent to a weighted L1-norm).

Note that Lipschitz-continuity is here always relative to the simplex $\Delta$, not $\mathbb{R}^{|\mathcal{S}|}$. $\Delta$ and $\mathcal{A}$ being both compact, properties that hold in the local and vector setting also hold in the uniform and/or scalar setting (but bounds are looser).

## 4 Lipschitz-continuity of $V^*$

Assuming a $\rho$-POMDP with local and vector Lipschitz-continuous reward function with "constant" $\boldsymbol{\lambda}_\rho(b, a)$ in $(b, a)$, this section first states that Bellman's optimality operator preserves the LC property, which then allows proving that $V^*$ is LC for finite horizons, but not necessarily for infinite ones.

**Proposition 1** ($\mathcal{H}$ **preserves Lipschitz-Continuity**). *Given a $\rho$-POMDP with $\boldsymbol{\lambda}_\rho(\cdot, \cdot)$-LC reward function, and a $\boldsymbol{\lambda}_V(\cdot)$-LC value function $V$, then $\mathcal{H}V$ is (at least) $\boldsymbol{\lambda}_{\mathcal{H}V}(\cdot)$-LC with, in each belief $b$,*

$$\boldsymbol{\lambda}_{\mathcal{H}V}(b) = \vec{\max}_a \Big[\boldsymbol{\lambda}_\rho(b, a) + \gamma \sum_z \big[(|V(b^{a,z})| + \boldsymbol{\lambda}_V(b^{a,z})b^{a,z}) \mathbf{1} + \boldsymbol{\lambda}_V(b^{a,z})\big] P_{a,z}\Big]. \quad (1)$$

**Proof** (sketch). A key point here is to show that $\kappa(\boldsymbol{w}) \doteq \|\boldsymbol{w}\|_1 V(\hat{\boldsymbol{w}})$ is LC (see supplementary material), which relies on the triangle (in)equality. Then the rest consists essentially in some algebra using this property and other LC properties. $\qquad\square$

As can be observed in Eq. (1), the resulting update formula of the value function's Lipschitz constant exploits both the locality (dependence on the belief $b$) and the use of a vector rather than a scalar.

The dependence of the update formula on $|V(b^{a,z})|$ may seem surprising since adding constant $k_r \in \mathbb{R}$ to $\rho(b, a)$ should induce adding a related constant $k_V$ to $V$ without changing local Lipschitz constants. This dependence is due to approximations made in the proof. $|V(b^{a,z})|$ can in fact be replaced by $|V(b^{a,z}) + k_V|$ in Equation 1 with a tunable $k_V$. This induces the multi-objective task of minimizing the components of the (vector) Lipschitz constant through $k_V$, a problem we address by setting $k_V = -\frac{\max_z V(b^{a,z}) + \min_z V(b^{a,z})}{2}$.

**Theorem 1 (Local Lipschitz-continuity of $V^*$ for finite $T$).** *Given a $\rho$-POMDP with $\boldsymbol{\lambda}_\rho(b, a)$-LC $\rho$, for any finite time horizon $T$, the optimal value function is (locally+vector) LC.*

**Proof.** The value function for $T = 0$ is trivially 0-LC.

By induction, as Bellman's optimality operator preserves the LC property (Prop. 1), the optimal value function is LC for any finite $T$. $\qquad\square$

**Asymptotic Behavior**   Previous results do not tell whether the resulting Lipschitz constant tends to a limit value when $T$ goes to infinity. This issue is considered here for a *scalar and uniform* constant $\lambda$. $\boldsymbol{\lambda}_\rho(b, a)$ and $\boldsymbol{\lambda}_t(b)$ (the Lipschitz constant for $V_t$, not indicating the horizon $T$) respectively become $\lambda_\rho$ and $\lambda_t$.

**Corollary 1.** *For a $\rho$-POMDP with (uniform) $\lambda_\rho$-LC $\rho$, the optimal value function verifies, for all $t$ and all $b_1, b_2$,*

$$|V_t^*(b_1) - V_t^*(b_2)| \leq \lambda_t \|b_1 - b_2\|_1, \; \textit{where } \lambda_t = \begin{cases} \left(\lambda_\rho + \gamma V^{lim}\right) \dfrac{1 - (2\gamma)^{T-t}}{1 - (2\gamma)} & \textit{if } \gamma \neq \frac{1}{2}, \\ \left(\lambda_\rho + \gamma V^{lim}\right)(T - t) & \textit{if } \gamma = \frac{1}{2}, \end{cases}$$

*with $V^{lim} \doteq \frac{1}{1-\gamma} \max_{b,a} |r(b, a)|$.*

In the common case $\gamma \geq \frac{1}{2}$, $\lambda_0$ diverges when $T \to \infty$. Hence, $V^*$ may not be LC in the infinite horizon setting. Yet, it will suffice to compute finite-horizon LC approximations of $V^*$ (as usual with PWLC approximations), as explained in the next section.

# 5   Approximating $V^*$

This section shows how to define, initialize, update and prune LC upper- and lower-bounding approximators of $V^*$, and then derive an $\epsilon$-optimal variant of the HSVI algorithm.

## 5.1   Upper- and Lower-Bounding $V^*$

The upper-bounding LC approximator is defined as a finite set of downward-pointing $L1$-cones (see Figure 1 (left)), where an upper-bounding cone $c_{\boldsymbol{\beta}}^U = \langle \boldsymbol{\beta}, u, \boldsymbol{\lambda} \rangle$—located at belief $\beta$, with "summit" value $u$ and "slope" vector $\boldsymbol{\lambda}$—induces a function $U_{\boldsymbol{\beta}}(b) = u + \boldsymbol{\lambda} \cdot |\boldsymbol{\beta} - b|$. The upper bound is thus defined as the lower envelope of a set of cones $C^U = \{c_{\boldsymbol{\beta}}^U\}_{\boldsymbol{\beta} \in \mathcal{B}^U}$—i.e., $U(b) = \min_{\boldsymbol{\beta} \in \mathcal{B}^U} U_{\boldsymbol{\beta}}(b)$. Respectively, for the lower-bounding approximator: a lower-bounding (upward-pointing) cone $c_{\boldsymbol{\beta}}^L = \langle \boldsymbol{\beta}, l, \boldsymbol{\lambda} \rangle$ induces a function $L_{\boldsymbol{\beta}}(b) = l - \boldsymbol{\lambda} \cdot |\boldsymbol{\beta} - b|$; and the lower bound is defined as the upper envelope of a set of cones $C^L = \{c_{\boldsymbol{\beta}}^L\}_{\boldsymbol{\beta} \in \mathcal{B}^L}$—i.e., $L(b) = \max_{\boldsymbol{\beta} \in \mathcal{B}^L} L_{\boldsymbol{\beta}}(b)$.

We now (i) show how the (pointwise) update of the upper- or lower-bound preserves this representation; (ii) verify that the properties required for HSVI to converge to an $\epsilon$-optimal solution still hold; and (iii) discuss their initialization.

**Updating (Upper and Lower) Bounds**   The following proposition and its counterpart state that, for both $U$ and $L$, a pointwise update results in adding a new cone with its own Lipschitz constant.

**Theorem 2 (Updating $U$).** *Let us assume that (i) $\rho$ is $\boldsymbol{\lambda}_\rho(b, a)$-LC for each $(b, a)$, and (ii) the upper bound $U$ is described by a set of upper cones $C^U$. Then, for any $b$, an improved upper bound is obtained by adding a cone in $b$, with value and Lipschitz constant:*

$$u(b) = [\mathcal{H}U](b) = \max_a \left( \rho(b, a) + \gamma \sum_z \|P_{a,z} b\|_1 U(b^{a,z}) \right),$$

$$\boldsymbol{\lambda}(b) = \vec{\max}_{a'} \left( \boldsymbol{\lambda}_\rho(b, a') + \gamma \sum_z \left[ \boldsymbol{\lambda}(\boldsymbol{\beta}^{a',z}) + \left( |u(\boldsymbol{\beta}^{a',z})| + \boldsymbol{\lambda}(\boldsymbol{\beta}^{a',z})\boldsymbol{\beta}^{a',z} \right) \mathbf{1} \right] P_{a',z} \right),$$

*where $\boldsymbol{\beta}^{a,z}$ is the current point in $\mathcal{B}'$ that best approximates $U$ in $b^{a,z}$ (see Fig. 1 (left)).*

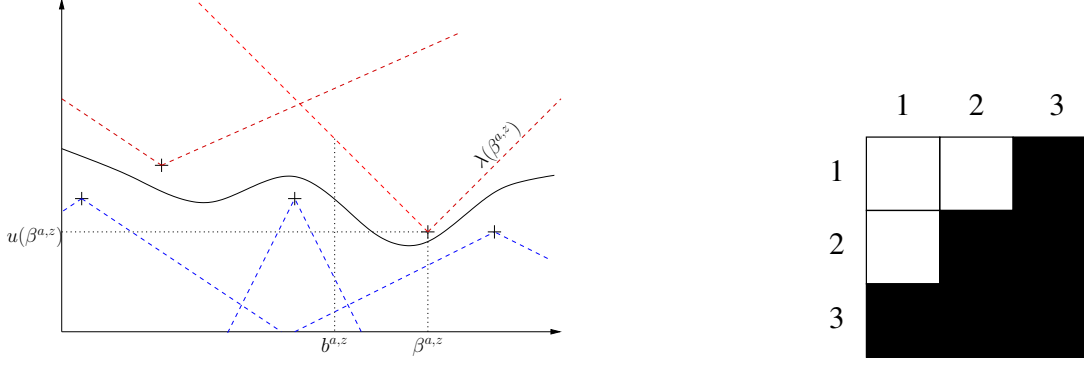

Figure 1: (left) An optimal $V^*$ surrounded by its upper and lower bounds (2 cones for $U$ in red, and 3 cones for $L$ in blue). The value of $U$ at $b^{a,z}$ is approximated by the cone located at $\boldsymbol{\beta}^{a,z}$. (right) Grid environment of the `grid-info` problem (Sec. 6.1).

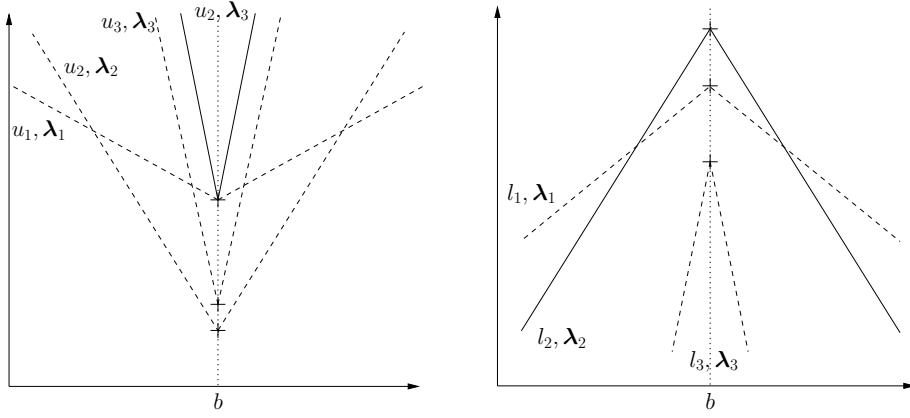

Figure 2: Bellman update of $U$ (left) and $L$ (right) at $b$ resulting in 1 (dashed) cone per action. (right) The upper envelope of 3 cones is approximated by (solid) cone $\langle b, u_2, \lambda_3 \rangle$. (left) The upper envelope of 3 cones is approximated by (solid) cone $\langle b, l_2, \lambda_2 \rangle$.

The operator performing this update at $b$ is noted $K_b^U$ and the updated upper bound is thus $K_b^U U$. Intuitively (see Fig. 2 (left)), each action induces one cone, which may be preferred depending on the point $b'$ where the evaluation is done. Yet, rather than adding the upper enveloppe of this set of cones, a single upper-bounding cone is employed using the maximum $u(b)$ and $\boldsymbol{\lambda}(b)$.

A pointwise update of $U$ in $b$ consists in computing (i) $u(b)$ by Bellman update, in $\Theta(|\mathcal{A}| \times |\mathcal{Z}| \times (|\mathcal{S}|^2 + |\mathcal{B}^U| \times |\mathcal{S}|))$ (memoizing points $\boldsymbol{\beta}^{a,z}$ that optimize $U(b^{a,z})$) and (ii) $\boldsymbol{\lambda}(b)$ by a Bellman-like update, in $\Theta(|\mathcal{A}| \times |\mathcal{Z}| \times |\mathcal{S}|^2)$. The latter computation searches for the worst Lipschitz constant component by component, thus harming the generalization capabilities. Better Lipschitz constants could be obtained by first pruning cones that are dominated by other cones.

For the lower bound $L$, an update requires adding at least one of the cones with maximum $l$, but we even add each cone induced by each action. The complexity is unchanged (replacing $\mathcal{B}^U$ by $\mathcal{B}^L$).

**Pruning Cones** The LC setting requires a procedure for pruning cones. Without loss of generality, we consider $U$. A cone $c_{\boldsymbol{\beta}}^U$ must be maintained if there exists $b \in \Delta$ such that $U_{\boldsymbol{\beta}}(b) < \min_{\boldsymbol{\beta}' \in \mathcal{B}^U \setminus \{\boldsymbol{\beta}\}} U_{\boldsymbol{\beta}'}(b)$, which is equivalent to finding a strictly negative value of $\phi_{\boldsymbol{\beta}}^U(b) \doteq U_{\boldsymbol{\beta}}(b) - \min_{\boldsymbol{\beta}' \in \mathcal{B}^U \setminus \{\boldsymbol{\beta}\}} U_{\boldsymbol{\beta}'}(b)$. This could be done by applying a minimization procedure on $\phi_{\boldsymbol{\beta}}^U$ until a negative value is found. A more pragmatic approach is to search, for cone $c_{\boldsymbol{\beta}}^U$, if another cone $c_{\boldsymbol{\beta}'}^U$ exists that completely dominates it—*i.e.*, (i) $u(\boldsymbol{\beta}) \geq u(\boldsymbol{\beta}') + \boldsymbol{\lambda}(\boldsymbol{\beta}')|\boldsymbol{\beta} - \boldsymbol{\beta}'|$, and (ii) $\boldsymbol{\lambda}(\boldsymbol{\beta}) \geq \boldsymbol{\lambda}(\boldsymbol{\beta}')$. This can be improved by comparing not the Lipschitz constants, but the value at

the corners of the simplex, to make sure that dominance is checked *inside* $\Delta$. The resulting process is conservative and cheap, but not complete.

**Preservation of HSVI's Convergence Properties**  Ensuring finite time convergence of HSVI (even beyond POMDPs) to an $\epsilon$-optimal solution requires using **(a)** a *uniformly improvable* (UI) lower bound $L$, *i.e.*, $\mathcal{H}L \geq L$, where $\mathcal{H}$ is Bellman optimality operator; **(b)** respectively a *uniformly improvable* (UI) upper bound $U$; **(c)** a *strong* pointwise update operator for the lower bound $L$, $K_{\cdot}^L$, *i.e.*, for each $b$ where it is applied and any $L$, (i) $(\mathcal{H}L)(b) = (K_b^L L)(b)$ and (ii) $(\mathcal{H}L)(b') \geq (K_b^L L)(b')$ in any other point $b'$; and **(d)** resp. a *strong* pointwise update operator for $U$, $K_{\cdot}^U$.

Trivially, our proposed operators are strong, thus conservative [Smith, 2007, Def. 3.24, 3.25]. Also, any conservative update operator preserves UI [Smith, 2007, Th. 3.29]. We thus essentially need to ensure that initializations induce UI bounds (see next sub-section).

**Initialization**  For a usual POMDP ($\rho = r$), initializations described by Smith [2007] are UI by construction. For a $\rho$-POMDP, similar constructions seem difficult to obtain. Another option is to go back to a POMDP with reward (linear in $b$) $r_u$ upper-bounding (resp. $r_l$ lower-bounding) $\rho$. We can then (i) employ the associated POMDP initialization, or (ii) solve the resulting POMDPs. In each case, the resulting bounds can be used as UI LC (with infinitely many cones). Going further, $\rho$ could even be better upper- (resp. lower-) bounded by a lower (resp. upper) envelope of linear reward functions, which would lead to better initializations of $U$ (resp. $L$) by taking lower (resp. upper) envelopes of independent bounds.

## 5.2  Algorithms

We will distinguish HSVI variants depending on the approximators at hand: pwlc-HSVI, lc-HSVI and pw-HSVI respectively depend on the classical PWLC approximators, the LC approximators previously described, and non-generalizing pointwise (PW) approximators (equivalent to cones with an infinite Lipschitz constant). In each case, HSVI's convergence guarantees hold.

**Incremental Variant**  lc-HSVI computes (using Eq.(1)) upper bounds on the *true* (local and vector) Lipschitz constants—i.e., the smallest constants for which the Lipschitz property holds. Yet, these upper bounds are often very pessimistic, which leads to (i) a poor generalization capability of $U$ and $L$, and, (ii) as a consequence, a very slow convergence. To circumvent the resulting pessimistic bounds and to assess how much is lost due to this pessimism, we also propose another algorithm that incrementally searches for a valid (global and scalar) Lipschitz constant $\lambda$. The intuition is that, despite the search process, the resulting planning process could be more efficient due to (i) quickly detecting when a constant is invalid, and (ii) quickly converging to a solution when a valid constant is found. One issue is that the algorithm may terminate with an invalid solution.

We first need to define lc-HSVI($\lambda$), a variant of lc-HSVI where the Lipschitz constant is uniformly constrained to (scalar) value $\lambda$. As a consequence, (i) adding a new cone at $\boldsymbol{\beta}$ only requires computing $u(\boldsymbol{\beta})$ or $l(\boldsymbol{\beta})$, and (ii) the pragmatic pruning process is complete. If $\lambda$ is not large enough, the algorithm may fail due to (LXU) $L$ and $U$ *crossing* each other at an update point $b$, (NUI) $L$ or $U$ being *not uniformly improvable*—i.e., an update leads to a worse value than expected—or, (UR) *unstable results*—i.e., two consecutive runs with values $\lambda_t$ and $\lambda_{t+1}$ verify $|L_t(b_0) - L_{t+1}(b_0)| > \epsilon$.

Then, lc-HSVI($\boldsymbol{\lambda}$) is incorporated in an incremental algorithm, inc-lc-HSVI (see Alg. 1, fct. `inc-lc-HSVI`), that starts with some initial $\lambda$ and runs lc-HSVI($\lambda$) with geometrically increasing values of $\lambda$ until lc-HSVI($\lambda$) returns with no (LXU/NUI/UR) failure. As already mentioned, this process does not guarantee that a large enough $\lambda$ has been found for $L$ and $U$ to be proper bounds. In practice, we use $\lambda_0 = 1$, but problem-dependent values should be considered to avoid being sensitive to affine transforms of the reward function.

# 6 Experiments

## 6.1 Benchmark Problems

To evaluate the various algorithms at hand, we consider both POMDP and $\rho$-POMDP benchmark problems. The former problems—a diverse set taken from Cassandra's POMDP page[3]—allow comparing the proposed algorithms against the standard pwlc-HSVI. The $\rho$-POMDP problems are all based on a grid environment as described below.

**grid-info $\rho$-POMDP** We consider an agent moving on a $3 \times 3$ toric grid with white and black cells (see Fig. 1 (right)). Each cell is indicated by its coordinates $(x, y)$ ($\in \{1, 2, 3\}$). The agent is initially placed uniformly at random. Moves (n,s,e,w) succeed with probability .8, otherwise the agent stays still. The current cell's color is observed with probability 1. $\gamma$ is set to $0.95$.

Let $b_x$ (resp. $b_y$) be the belief over the $x$ (resp. $y$) coordinate. Then, $\rho(b) = +\|b_x - \frac{1}{3}\mathbf{1}\|_1$ (resp. $-\|b_x - \frac{1}{3}\mathbf{1}\|_1$) rewards *knowing $x$* (k$x$) (resp. *not knowing $x$* (¬k$x$)). And replacing $b_x$ by $b_y$ allows rewarding *knowing $y$* (k$y$) and *not knowing $y$* (¬k$y$).

## 6.2 Experiments

We run $x$-HSVI ($x \in \{\text{pwlc}, \text{pw}, \text{lc}, \text{inc-lc}\}$) on all benchmark problems—with the exception of pwlc-HSVI not being run on $\rho$-POMDPs—setting $\epsilon = 0.1$ and a timeout of 600s. In inc-lc-HSVI, $\boldsymbol{\lambda}$ is initially set to 1. $L$ and $U$ are initialized (i) for POMDPs, using HSVI1's blind estimate and MDP estimate, and (ii) for $\rho$-POMDPs, using $\frac{R_{min}}{1-\gamma}$ and $\frac{R_{max}}{1-\gamma}$. The Java program[4] is run on an i5 CPU M540 at 2.53GHz. Experimental results are presented in Table 1. When convergence is not achieved, we look at the final $L(b_0)$ and $U(b_0)$ values to assess the efficiency of an algorithm. Note that, for inc-lc-HSVI, $\log_2(\boldsymbol{\lambda})$ gives the number of restarts.

**inc-lc-HSVI's Restart Criteria** We first look at the effect of the three restart criteria in inc-lc-HSVI through the top sub-table. The first two columns are similar, showing that not testing that $L$ and $U$ cross each other (noLXU) has little influence. Looking at execution traces, the LXU criterion is in fact only triggered when not checking for uniform improvability (noNUI). The time to converge is notably sped up by not testing for unstable results (noUR), with only one case of convergence to bad values in the Tiger problem (tiger70). More speed improvement is obtained by not testing uniform improvability (noNUI), in which case the LXU rule is triggered more often. As a result, we take as our default configuration the "noNUI" setting—which only uses the LXU and UR stopping criteria.

**Comparing Approximators and Algorithms** We now compare the four algorithms at hand through the bottom sub-table. pwlc-HSVI (when applicable) dominates overall the experiments, except on a few cases where inc-lc-HSVI converges in less time. As can be observed, the Lipschitz constants obtained by inc-lc-HSVI are of the same order of magnitude as the ones derived in pwlc-HSVI from the final lower bounds $L$. inc-lc-HSVI(noNUI) would be a satisfying solution on the benchmarks at hand (when not using the PWLC property) if not lacking theoretical guarantees. For its part, lc-HSVI ends up with worst-case constants orders of magnitude larger in many cases, which suggests that its bounds have little generalization capabilities, as in pw-HSVI. pw-HSVI is obviously faster than lc-HSVI due to much cheaper updates.

# 7 Discussion

This work shows that, for finite horizons, the optimal value function of a $\rho$-POMDP with Lipschitz $\rho$ is Lipschitz-continuous (LC). The Lipschitz-continuity here is not uniform with a scalar constant, but local with a vector constant, which allows for more efficient updates. While the PWLC property (of $V^*$) provides useful generalizing lower and upper bounds for POMDPs and $\rho$-POMDPs with convex $\rho$, the LC property provides similar bounds for POMDPs and $\rho$-POMDPs with LC $\rho$—where $V^*$ may not be convex. These bounds are envelopes of either upward- or downward-pointing "cones", and,

Table 1: Comparison of (top) inc-lc-HSVI with all 3 stopping criteria, or with 1 of them disabled and (bottom) $x$-HSVI algorithms (for $x \in \{\text{pwlc}, \text{pw}, \text{lc}, \text{inc-lc}\}$), in terms of (i) CPU time (timeout 600s), (ii) number of trajectories, (iii-iv) width (gap) at $b_0$, and (v) Lipschitz constant

| $x$-HSVI | inc-lc | | | | inc-lc(noLXU) | | | | inc-lc(noNUI) | | | | inc-lc(noUR) | | | |
|---|---|---|---|---|---|---|---|---|---|---|---|---|---|---|---|---|
| | $t$ (s) | (#it) | $gap(b_0)$ | $\lambda$ | $t$ (s) | (#it) | $gap(b_0)$ | $\lambda$ | $t$ (s) | (#it) | $gap(b_0)$ | $\lambda$ | $t$ (s) | (#it) | $gap(b_0)$ | $\lambda$ |
| 4x3.95 | **1** | (254) | 0.10 | 1 | **1** | (254) | 0.10 | 1 | **1** | (254) | 0.10 | 1 | **1** | (254) | 0.10 | 1 |
| 4x4.95 | **0** | (125) | 0.10 | 1 | **0** | (125) | 0.10 | 1 | **0** | (125) | 0.10 | 1 | **0** | (125) | 0.10 | 1 |
| cheese.95 | **0** | (69) | 0.10 | 1 | **0** | (69) | 0.10 | 1 | **0** | (69) | 0.10 | 1 | **0** | (69) | 0.10 | 1 |
| cit | 600 | (44) | 0.67 | 1 | 600 | (48) | 0.67 | 1 | 600 | (41) | 0.67 | 1 | 600 | (44) | 0.67 | 1 |
| hallway | 600 | (645) | 1.11 | 1 | 600 | (642) | 1.11 | 1 | 600 | (611) | 1.12 | 1 | 600 | (622) | 1.12 | 1 |
| hallway2 | 600 | (687) | 1.00 | 1 | 600 | (686) | 1.00 | 1 | 600 | (668) | 1.00 | 1 | 600 | (694) | 1.00 | 1 |
| milos-aaai97 | 600 | (1331) | 49.25 | 512 | 600 | (1300) | 49.45 | 512 | 600 | (1797) | 43.78 | 64 | 600 | (1287) | 49.47 | 512 |
| mit | 600 | (46) | 0.69 | 1 | 600 | (47) | 0.69 | 1 | 600 | (47) | 0.69 | 1 | 600 | (46) | 0.69 | 1 |
| network | 600 | (8572) | 0.44 | 512 | 600 | (8462) | 0.47 | 512 | 34 | (2819) | 0.10 | 128 | 258 | (6849) | 0.10 | 256 |
| paint.95 | **0** | (84) | 0.10 | 1 | **0** | (84) | 0.10 | 1 | **0** | (84) | 0.10 | 1 | **0** | (84) | 0.10 | 1 |
| pentagon | 600 | (62) | 0.73 | 1 | 600 | (56) | 0.73 | 1 | 600 | (60) | 0.73 | 1 | 600 | (56) | 0.73 | 1 |
| shuttle.95 | **0** | (47) | 0.08 | 4 | **0** | (47) | 0.08 | 4 | **0** | (47) | 0.08 | 4 | **0** | (47) | 0.08 | 4 |
| tiger85 | **0** | (15) | 0.07 | 64 | **0** | (15) | 0.07 | 64 | **0** | (15) | 0.07 | 64 | **0** | (16) | 0.07 | 32 |
| tiger-grid | 600 | (455) | 3.95 | 8 | 600 | (449) | 3.95 | 8 | 600 | (626) | 0.78 | 1 | 600 | (456) | 3.95 | 8 |
| grid-info k$x$ | 12 | (242) | 0.95 | 64 | 12 | (242) | 0.95 | 64 | 1 | (279) | 0.10 | 4 | 8 | (350) | 0.10 | 32 |
| grid-info k$y$ | 27 | (395) | 0.10 | 512 | 26 | (395) | 0.10 | 512 | 1 | (279) | 0.10 | 4 | 26 | (395) | 0.10 | 512 |
| grid-info ¬k$x$ | 600 | (1482) | 0.83 | 128 | 600 | (1535) | 0.72 | 128 | 4 | (695) | 0.10 | 4 | 151 | (1572) | 0.10 | 32 |
| grid-info ¬k$y$ | 600 | (757) | 9.19 | 2048 | 600 | (717) | 9.40 | 2048 | 4 | (707) | 0.10 | 4 | 600 | (773) | 9.14 | 2048 |

| $x$-HSVI | pwlc | | | | pw | | | lc | | | | inc-lc(noNUI) | | | |
|---|---|---|---|---|---|---|---|---|---|---|---|---|---|---|---|
| | $t$ (s) | (#it) | $gap(b_0)$ | $\lambda$ | $t$ (s) | (#it) | $gap(b_0)$ | $t$ (s) | (#it) | $gap(b_0)$ | $\lambda$ | $t$ (s) | (#it) | $gap(b_0)$ | $\lambda$ |
| 4x3.95 | **1** | (134) | 0.10 | 1.19 | 600 | (447) | 0.94 | 600 | (214) | 3.27 | 1.9e+05 | **1** | (254) | 0.10 | 1 |
| 4x4.95 | **1** | (120) | 0.10 | 0.66 | **1** | (134) | 0.10 | 6 | (134) | 0.10 | 2.4e+05 | **0** | (125) | 0.10 | 1 |
| cheese.95 | **0** | (59) | 0.10 | 1.15 | **0** | (69) | 0.10 | 1 | (69) | 0.10 | 4.8e+04 | **0** | (69) | 0.10 | 1 |
| cit | 600 | (19) | 0.13 | 1.39 | 600 | (126) | 0.84 | 601 | (34) | 0.84 | 3.0e+01 | 600 | (41) | 0.67 | 1 |
| hallway | 600 | (414) | 0.35 | 0.70 | 600 | (683) | 1.30 | 600 | (203) | 1.33 | 1.8e+00 | 600 | (611) | 1.12 | 1 |
| hallway2 | 600 | (385) | 0.67 | 0.63 | 600 | (690) | 1.04 | 600 | (208) | 1.06 | 4.0e+00 | 600 | (668) | 1.00 | 1 |
| milos-aaai97 | 600 | (1152) | 29.55 | 89.49 | 600 | (1725) | 49.05 | 600 | (595) | 52.87 | 2.3e+03 | 600 | (1797) | 43.78 | 64 |
| mit | 600 | (21) | 0.12 | 1.81 | 600 | (236) | 0.87 | 600 | (64) | 0.88 | 1.9e+17 | 600 | (47) | 0.69 | 1 |
| network | 498 | (7703) | 0.10 | 168.22 | 600 | (3021) | 453.62 | 600 | (941) | 510.76 | 5.0e+16 | 34 | (2819) | 0.10 | 128 |
| paint.95 | 2 | (143) | 0.10 | 1.00 | 600 | (3695) | 3.55 | 600 | (1008) | 4.37 | 1.7e+281 | **0** | (84) | 0.10 | 1 |
| pentagon | 601 | (27) | 0.31 | 1.00 | 600 | (89) | 0.83 | 600 | (32) | 0.83 | 2.8e+01 | 600 | (60) | 0.73 | 1 |
| shuttle.95 | **0** | (23) | 0.10 | 22.77 | **0** | (42) | 0.09 | **0** | (42) | 0.09 | 7.5e+00 | **0** | (47) | 0.08 | 4 |
| tiger85 | **0** | (15) | 0.09 | 55.00 | **0** | (15) | 0.08 | **0** | (15) | 0.08 | 2.2e+02 | **0** | (15) | 0.07 | 64 |
| tiger-grid | 600 | (264) | 0.51 | 11.45 | 600 | (1563) | 16.74 | 600 | (375) | 17.23 | 4.5e+01 | 600 | (626) | 0.78 | 1 |
| grid-info k$x$ | – | (–) | – | – | 600 | (709) | 0.24 | 600 | (358) | 1.82 | 3.6e+01 | **1** | (279) | 0.10 | 4 |
| grid-info k$y$ | – | (–) | – | – | 27 | (432) | 0.10 | 344 | (426) | 0.10 | 4.0e+01 | **1** | (279) | 0.10 | 4 |
| grid-info ¬k$x$ | – | (–) | – | – | 600 | (2319) | 9.15 | 600 | (889) | 13.74 | 3.3e+01 | 4 | (695) | 0.10 | 4 |
| grid-info ¬k$y$ | – | (–) | – | – | 600 | (1393) | 6.66 | 600 | (604) | 8.52 | 3.1e+01 | 4 | (707) | 0.10 | 4 |

with appropriate initializations, are uniformly improvable. Two algorithms are proposed: HSVI used with these "LC bounds"—which preserves HSVI's convergence properties—, and an incremental algorithm that searches for a (uniform) scalar Lipschitz constant allowing for fast computations—with no guarantees that the bounds are valid.

The experiments show that there lc-HSVI's pessimistic constants are far from inc-lc-HSVI's guesses. This encourages searching for better (safe) Lipschitz constants—possibly using a particular norm such that the dynamics of the bMDP are LC,[5] as Platzman [1977] did for *sub-rectangular* bMDPs (a restrictive class of problems)—but also improving the initialization of $L$ and $U$, and possibly inc-lc-HSVI's restart and stopping criteria (ideally guaranteeing that a valid constant is found).

We also aim at exploiting the Lipschitz continuity to solve partially observable stochastic games (POSGs) [Hansen et al., 2004]. Indeed, while the PWLC property allows efficiently solving not only POMDPs, but also Dec-POMDPs turned into occupancy MDPs [Dibangoye et al., 2013, 2016], the LC property may allow to provide generalizing bounds for POSGs turned into occupancy SGs, starting with 2-player 0-sum scenarios.

## Acknowledgments

Let us thank David Reboullet for helping with some proofs (reminding us about the power of the triangle (in)equality), and the anonymous reviewers for their insightful comments.

## Footnotes

[1] And also to solving Decentralized POMDPs (Dec-POMDPs) as occupancy MDPs (oMDPs)—*i.e.*, when designing multiple collaborating controllers—[Dibangoye et al., 2013, 2016].

[2]It is thus also Lipschitz-continuous.

[3]http://www.pomdp.org/examples/

[4]Full code available here: https://gitlab.inria.fr/buffet/lc-hsvi-nips18

[5]Previously cited works on LC MDPs rely on LC dynamics.

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
