[Supplementary Material]

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

# A  Proofs (including preliminary results)

Let us first (i) remind that some operations on vectors or vector-valued functions are component-by-component operations, such as the absolute value $|\boldsymbol{v}|$, the m$\vec{\text{a}}$x operator, or comparisons such as $\vec{\leq}$; and (ii) underline that the Lipschitz continuity is local (and vector) everywhere but in Corollary 1.

## A.1  Proof of Proposition 1 ($\mathcal{H}$ preserves Lipschitz-Continuity)

Proving Proposition 1 requires first proving Lemmas 2 (which requires Lemma 1) and 3.

**Lemma 1.** *Let $\|\cdot\|$ be some norm. If $\boldsymbol{u}$ and $\boldsymbol{v}$ in $\mathbb{R}^n$ verify $\|\boldsymbol{u}\| > 0$ and $\|\boldsymbol{v}\| > 0$, then*

$$\|\boldsymbol{u}\|\left|\frac{\boldsymbol{u}}{\|\boldsymbol{u}\|} - \frac{\boldsymbol{v}}{\|\boldsymbol{v}\|}\right| \vec{\leq} |\boldsymbol{u} - \boldsymbol{v}| + \|\boldsymbol{u} - \boldsymbol{v}\|\frac{|\boldsymbol{v}|}{\|\boldsymbol{v}\|}.$$

**Proof.** The proof results from writing:

$$\|\boldsymbol{u}\|\left|\frac{\boldsymbol{u}}{\|\boldsymbol{u}\|} - \frac{\boldsymbol{v}}{\|\boldsymbol{v}\|}\right| \vec{\leq} \|\boldsymbol{u}\|\left|\frac{\boldsymbol{u}}{\|\boldsymbol{u}\|} - \frac{\boldsymbol{v}}{\|\boldsymbol{u}\|}\right| + \|\boldsymbol{u}\|\left|\frac{\boldsymbol{v}}{\|\boldsymbol{u}\|} - \frac{\boldsymbol{v}}{\|\boldsymbol{v}\|}\right| \quad \text{(by componentwise triangle inequality)}$$

$$= \|\boldsymbol{u}\|\left|\frac{\boldsymbol{u}}{\|\boldsymbol{u}\|} - \frac{\boldsymbol{v}}{\|\boldsymbol{u}\|}\right| + \left|\|\boldsymbol{v}\|\frac{\boldsymbol{v}}{\|\boldsymbol{v}\|} - \|\boldsymbol{u}\|\frac{\boldsymbol{v}}{\|\boldsymbol{v}\|}\right|$$

$$= |\boldsymbol{u} - \boldsymbol{v}| + \left|\|\boldsymbol{v}\| - \|\boldsymbol{u}\|\right|\frac{|\boldsymbol{v}|}{\|\boldsymbol{v}\|} \quad \text{(since the vectors in the second term are colinear)}$$

$$\vec{\leq} |\boldsymbol{u} - \boldsymbol{v}| + \|\boldsymbol{u} - \boldsymbol{v}\|\frac{|\boldsymbol{v}|}{\|\boldsymbol{v}\|} \quad \text{(by reverse triangle inequality)}.$$

$\square$

**Lemma 2.** *Let $f : \mathbb{R}^{+^n} \to \mathbb{R}$ be $\boldsymbol{\lambda}_f(\cdot)$-LC. Then $\kappa(\boldsymbol{w}) \doteq \|\boldsymbol{w}\|_1 f(\hat{\boldsymbol{w}})$ is $\boldsymbol{\lambda}_\kappa(\cdot)$-LC with, for all $\boldsymbol{w}$,*

$$\boldsymbol{\lambda}_\kappa(\boldsymbol{w}) \doteq [|f(\hat{\boldsymbol{w}})| + \boldsymbol{\lambda}_f(\hat{\boldsymbol{w}})\hat{\boldsymbol{w}}]\,\mathbf{1} + \boldsymbol{\lambda}_f(\hat{\boldsymbol{w}}).$$

**Proof.** For all non-zero $\boldsymbol{w}_1$ and $\boldsymbol{w}_2$, we have:

$$\kappa(\boldsymbol{w}_1) - \kappa(\boldsymbol{w}_2)$$

$$= \|\boldsymbol{w}_1\|_1 f(\frac{\boldsymbol{w}_1}{\|\boldsymbol{w}_1\|_1}) - \|\boldsymbol{w}_2\|_1 f(\frac{\boldsymbol{w}_2}{\|\boldsymbol{w}_2\|_1}) \quad \text{(by definition)}$$

$$\leq \|\boldsymbol{w}_1\|_1\left(f(\frac{\boldsymbol{w}_2}{\|\boldsymbol{w}_2\|_1}) + \boldsymbol{\lambda}_f(\frac{\boldsymbol{w}_2}{\|\boldsymbol{w}_2\|_1})|\frac{\boldsymbol{w}_2}{\|\boldsymbol{w}_2\|_1} - \frac{\boldsymbol{w}_1}{\|\boldsymbol{w}_1\|_1}|\right) - \|\boldsymbol{w}_2\|_1 f(\frac{\boldsymbol{w}_2}{\|\boldsymbol{w}_2\|_1}) \quad \text{(because } f \text{ is } \boldsymbol{\lambda}_f(\cdot)\text{-LC)}$$

$$= (\|\boldsymbol{w}_1\|_1 - \|\boldsymbol{w}_2\|_1)f(\frac{\boldsymbol{w}_2}{\|\boldsymbol{w}_2\|_1}) + \boldsymbol{\lambda}_f(\frac{\boldsymbol{w}_2}{\|\boldsymbol{w}_2\|_1})\|\boldsymbol{w}_1\|_1|\frac{\boldsymbol{w}_2}{\|\boldsymbol{w}_2\|_1} - \frac{\boldsymbol{w}_1}{\|\boldsymbol{w}_1\|_1}|$$

$$\leq |\|\boldsymbol{w}_1\|_1 - \|\boldsymbol{w}_2\|_1||f(\frac{\boldsymbol{w}_2}{\|\boldsymbol{w}_2\|_1})| + \boldsymbol{\lambda}_f(\frac{\boldsymbol{w}_2}{\|\boldsymbol{w}_2\|_1})\left(|\boldsymbol{w}_1 - \boldsymbol{w}_2| + \|\boldsymbol{w}_1 - \boldsymbol{w}_2\|_1\frac{|\boldsymbol{w}_2|}{\|\boldsymbol{w}_2\|_1}\right) \quad \text{(using Lemma 1)}$$

$$\leq |f(\frac{\boldsymbol{w}_2}{\|\boldsymbol{w}_2\|_1})|\|\boldsymbol{w}_1 - \boldsymbol{w}_2\|_1 + \boldsymbol{\lambda}_f(\frac{\boldsymbol{w}_2}{\|\boldsymbol{w}_2\|_1})\left(|\boldsymbol{w}_1 - \boldsymbol{w}_2| + \frac{|\boldsymbol{w}_2|}{\|\boldsymbol{w}_2\|_1}\|\boldsymbol{w}_1 - \boldsymbol{w}_2\|_1\right)$$

$$\text{(by reverse triangle inequality)}$$

$$= \left(\left(|f(\frac{\boldsymbol{w}_2}{\|\boldsymbol{w}_2\|_1})| + \boldsymbol{\lambda}_f(\frac{\boldsymbol{w}_2}{\|\boldsymbol{w}_2\|_1})\frac{|\boldsymbol{w}_2|}{\|\boldsymbol{w}_2\|_1}\right)\mathbf{1} + \boldsymbol{\lambda}_f(\frac{\boldsymbol{w}_2}{\|\boldsymbol{w}_2\|_1})\right)|\boldsymbol{w}_1 - \boldsymbol{w}_2| \text{(where } \mathbf{1} \text{ is a row vector of 1s).}$$

$\square$

**Lemma 3.** *For two belief states $b_1$ and $b_2$, and for all action-observation pairs $(a, z)$, we have:*

$$|P_{a,z}b_1 - P_{a,z}b_2| \leq P_{a,z}|b_1 - b_2|.$$

**Proof.** $P_{a,z}$ being non-negative, $P_{a,z}b_1$ and $P_{a,z}b_2$ are vectors of $\mathbb{R}^{+^n}$ as $b_1$ and $b_2$, hence, for any $s'$,

$$|P_{a,z}b_1 - P_{a,z}b_2|(s') = |\sum_s P_{a,z}^{s,s'}(b_1(s) - b_2(s))|$$

$$\leq \sum_s P_{a,z}^{s,s'}|b_1(s) - b_2(s)|$$

$$= \left(P_{a,z}|b_1 - b_2|\right)(s').$$

$\square$

With Lemmas 2 and 3, we can now prove Proposition 1.

**Proof.** Let $b_1$ and $b_2$ in $\mathcal{B}$ be two belief points. By setting $\kappa(\boldsymbol{w}) = \|\boldsymbol{w}\|_1 V(\frac{\boldsymbol{w}}{\|\boldsymbol{w}\|_1})$, Lemma 2 allows us to write:

$$\kappa(P_{a,z}b_1)$$

$$\leq \kappa(P_{a,z}b_2) + \left(\left[|V(\frac{P_{a,z}b_2}{\|P_{a,z}b_2\|_1})| + \boldsymbol{\lambda}_V(\frac{P_{a,z}b_2}{\|P_{a,z}b_2\|_1})\frac{P_{a,z}b_2}{\|P_{a,z}b_2\|_1}\right]\mathbf{1} + \boldsymbol{\lambda}_V(\frac{P_{a,z}b_2}{\|P_{a,z}b_2\|_1})\right)|P_{a,z}b_1 - P_{a,z}b_2|$$

$$\leq \kappa(P_{a,z}b_2) + \left([|V(b_2^{a,z})| + \boldsymbol{\lambda}_V(b_2^{a,z})b_2^{a,z}]\mathbf{1} + \boldsymbol{\lambda}_V(b_2^{a,z})\right)P_{a,z}|b_1 - b_2| \qquad \text{(because } b_2^{a,z} = \frac{P_{a,z}b_2}{\|P_{a,z}b_2\|_1}\text{).}$$

From there, we get:

$$\mathcal{H}V(b_1) = \max_a \left[\rho(b_1, a) + \gamma \sum_z \|P_{a,z}b_1\|_1 V(\frac{P_{a,z}b_1}{\|P_{a,z}b_1\|_1})\right]$$

$$\leq \max_a \left[(\rho(b_2, a) + \boldsymbol{\lambda}_r(b_2, a)|b_1 - b_2|) + \gamma \sum_z \kappa(P_{a,z}b_1)\right] \quad \text{(by local Lipschitz-continuity of } \rho)$$

$$\leq \max_a \left[(\rho(b_2, a) + \boldsymbol{\lambda}_r(b_2, a)|b_1 - b_2|) + \gamma \sum_z \Big(\kappa(P_{a,z}b_2)\right.$$

$$\left. + \Big((|V(b_2^{a,z})| + \boldsymbol{\lambda}_V(b_2^{a,z})b_2^{a,z})\mathbf{1} + \boldsymbol{\lambda}_V(b_2^{a,z})\Big)P_{a,z}|b_1 - b_2|\Big)\right]$$

$$\text{(by local Lipschitz-continuity of } \kappa)$$

$$\leq \max_a \left[\rho(b_2, a) + \gamma \sum_z \kappa(P_{a,z}b_2)\right] \tag{2}$$

$$+ \max_a \left[\boldsymbol{\lambda}_r(b_2, a)|b_1 - b_2| + \gamma \sum_z ((|V(b_2^{a,z})| + \boldsymbol{\lambda}_V(b_2^{a,z})b_2^{a,z})\mathbf{1} + \boldsymbol{\lambda}_V(b_2^{a,z}))P_{a,z}|b_1 - b_2|\right]$$

then, by getting $|b_1 - b_2|$ out and by using a component-by-component $\max$,

$$\leq \underbrace{\max_a \left[\rho(b_2, a) + \gamma \sum_z \kappa(P_{a,z}b_2)\right]}_{\mathcal{H}V(b_2)}$$

$$+ \underbrace{\vec{\max}_a \left[\boldsymbol{\lambda}_r(b_2, a) + \gamma \sum_z ((|V(b_2^{a,z})| + \boldsymbol{\lambda}_V(b_2^{a,z})b_2^{a,z})\mathbf{1} + \boldsymbol{\lambda}_V(b_2^{a,z}))P_{a,z}\right]|b_1 - b_2|}_{\boldsymbol{\lambda}_{\mathcal{H}V}(b_2)}$$

$$= \mathcal{H}V(b_2) + \boldsymbol{\lambda}_{\mathcal{H}V}(b_2)|b_1 - b_2|.$$

Also, symmetrically:

$$\mathcal{H}V(b_1) = \max_a \left[\rho(b_1, a) + \gamma \sum_z \|P_{a,z}b_1\|_1 V(\frac{P_{a,z}b_1}{\|P_{a,z}b_1\|_1})\right]$$

$$\geq \max_a \left[(\rho(b_2, a) - \boldsymbol{\lambda}_r(b_2, a)|b_1 - b_2|) + \gamma \sum_z \kappa(P_{a,z}b_1)\right] \text{(by local Lipschitz-continuity of } \rho)$$

$$\geq \max_a \left[(\rho(b_2, a) - \boldsymbol{\lambda}_r(b_2, a)|b_1 - b_2|) + \gamma \sum_z \Big(\kappa(P_{a,z}b_2)\right.$$

$$\left. - \Big((|V(b_2^{a,z})| + \boldsymbol{\lambda}_V(b_2^{a,z})b_2^{a,z})\mathbf{1} + \boldsymbol{\lambda}_t(b_2^{a,z})\Big)P_{a,z}|b_1 - b_2|\Big)\right]$$

(by local Lipschitz-continuity of $\kappa$)

then, by getting $|b_1 - b_2|$ out and by using a component-by-component $\max$,

$$\geq \underbrace{\max_a \left[ \rho(b_2, a) + \gamma \sum_z \kappa(P_{a,z} b_2) \right]}_{\mathcal{H}V(b_2)}$$

$$- \underbrace{\vec{\max}_a \left[ \boldsymbol{\lambda}_r(b_2, a) + \gamma \sum_z \left( (|V(b_2^{a,z})| + \boldsymbol{\lambda}_V(b_2^{a,z}) b_2^{a,z}) \mathbf{1} + \boldsymbol{\lambda}_V(b_2^{a,z}) \right) P_{a,z} \right] |b_1 - b_2|}_{\boldsymbol{\lambda}_{\mathcal{H}V}(b_2)}$$

$$= \mathcal{H}V(b_2) - \boldsymbol{\lambda}_{\mathcal{H}V}(b_2) |b_1 - b_2|.$$

The value function $\mathcal{H}V$ is thus $\boldsymbol{\lambda}_{\mathcal{H}V}(b)$-Lipschitz-continuous with

$$\boldsymbol{\lambda}_{\mathcal{H}V}(b) =$$
$$\max_a \left[ \boldsymbol{\lambda}_r(b, a) + \gamma \sum_z \left( \boldsymbol{\lambda}_V(\frac{P_{a,z}b}{\|P_{a,z}b\|_1}) + \left( |V(\frac{P_{a,z}b}{\|P_{a,z}b\|_1})| + \boldsymbol{\lambda}_V(\frac{P_{a,z}b}{\|P_{a,z}b\|_1}) \frac{P_{a,z}b}{\|P_{a,z}b\|_1} \right) \mathbf{1} \right) P_{a,z} \right].$$

$\square$

### A.2 Proof of Corollary 1 (asymptotic behavior of the value function)

Note that the proof of this corollary starts with formulas using the local (and vector) Lipschitz continuity and ends with formulas using the uniform (and scalar) Lipschitz continuity.

**Proof.** Starting from Formula 2 in the proof of Prop. 1 we have, for some value function $V$,

$$\mathcal{H}V(b_1) - \mathcal{H}V(b_2)$$
$$\leq \max_a \left[ \left( \boldsymbol{\lambda}_r(b_2, a) + \gamma \sum_z \left( \boldsymbol{\lambda}_V(b_2^{a,z}) + (|V(b_2^{a,z})| + \boldsymbol{\lambda}_V(b_2^{a,z}) b_2^{a,z}) \mathbf{1} \right) P_{a,z} \right) |b_1 - b_2| \right]$$
$$\leq \max_a \left[ \boldsymbol{\lambda}_r |b_1 - b_2| + \gamma \sum_z \left( \boldsymbol{\lambda}_V + \left( \max_b |V_t^*(b)| + \boldsymbol{\lambda}_V b_2^{a,z} \right) \mathbf{1} \right) P_{a,z} |b_1 - b_2| \right]$$
$$= \lambda_r \|b_1 - b_2\|_1 + \max_a \left[ \gamma \sum_z \left( \boldsymbol{\lambda}_V + V^{lim} + \boldsymbol{\lambda}_V \underbrace{\mathbf{1} b_2^{a,z}}_{=\|b_2^{a,z}\|_1 = 1} \right) \mathbf{1} P_{a,z} |b_1 - b_2| \right]$$
$$= \lambda_r \|b_1 - b_2\|_1 + \max_a \left[ \gamma \left( V^{lim} + 2\lambda_V \right) \underbrace{\sum_z \mathbf{1} P_{a,z} |b_1 - b_2|}_{=\|b_1 - b_2\|_1} \right]$$
$$= \underbrace{\left( \lambda_r + \gamma \left( V^{lim} + 2\lambda_V \right) \right)}_{=\lambda_{\mathcal{H}V}} \|b_1 - b_2\|_1.$$

Let us now consider a problem with finite temporal horizon $T$ and the application of the above relation to the optimal value function—which is time-dependent. This relation *a priori* involves a different $V_t^{lim}$ at each time step, but it can easily be upper-bounded by a constant $V^{*,lim}$. Doing so, we get a sequence of Lipschitz constants (one per time step) defined by a first-order linear non-homogeneous recurrence with constant coefficients:

$$\lambda_T = 0 \quad \text{and, for each } t \in \{1, \ldots, T\},$$
$$\lambda_{t-1} = \underbrace{(2\gamma)}_{\alpha} \lambda_t + \underbrace{\left( \lambda_r + \gamma V^{*,lim} \right)}_{\beta}.$$

The solution of this sequence immediately leads to the expected result, which depends on $\alpha$ ($> 0$) being equal to 1 or not. $\square$

## A.3 Proof of Theorem 2 (updating (upper and lower) bounds)

Below is the proof of Theorem 2.

**Proof.** The description of upper-bound $U$ by a set of upper cones amounts to writing that, for all $b$,

$$U(b) = \min_{\beta \in \mathcal{B}^U} \underbrace{(u_\beta + \boldsymbol{\lambda}_\beta |b - \beta|)}_{\dot{=} U_\beta(b)}.$$

Yet, a Bellman update at a point $b$ gives:

$$K_b^U(b) = \max_a \left( \rho(b, a) + \gamma \sum_z \|P_{a,z} b\|_1 U(b^{a,z}) \right).$$

Through $U(b^{a,z})$, this formula involves a cone $\langle \beta^{a,z}, u_{\beta^{a,z}}, \boldsymbol{\lambda}_{\beta^{a,z}} \rangle$ for each $z$, where $\beta^{a,z} = \arg\min_{\beta \in \mathcal{B}^U} (u_\beta + \boldsymbol{\lambda}_\beta |b^{a,z} - \beta|)$. As, for each $z$, the cone $\langle b^{a,z}, U(b^{a,z}), \boldsymbol{\lambda}_{\beta^{a,z}} \rangle$ is included in (dominated by) the cone $\langle \beta^{a,z}, u_{\beta^{a,z}}, \boldsymbol{\lambda}_{\beta^{a,z}} \rangle$, we can write, noting $a_b$ the maximizing action:

$$K_b^U(b) = \rho(b, a_b) + \gamma \sum_z \|P_{a_b,z} b\|_1 \underbrace{[u_{\beta^{a_b,z}} + \boldsymbol{\lambda}_{\beta^{a_b,z}} |b^{a_b,z} - \beta^{a_b,z}|]}_{\dot{=} U_{\beta^{a_b,z}}(b^{a_b,z})}.$$

Applying Lemma 2 to function $\kappa$ defined as $\kappa(\boldsymbol{w}) = \|\boldsymbol{w}\|_1 U_{\beta^{a_b,z}}(\frac{\boldsymbol{w}}{\|\boldsymbol{w}\|_1})$, one also observes that, for another point $\hat{b} \in \mathcal{B}$,

$$\|P_{a_b,z} b\|_1 U_{\beta^{a_b,z}}(b^{a_b,z}) - \|P_{a_b,z} \hat{b}\|_1 U_{\beta^{a_b,z}}(\hat{b}^{a_b,z})$$
$$\leq \underbrace{([|u_{\beta^{a_b,z}}| + \boldsymbol{\lambda}_{\beta^{a_b,z}} \beta^{a_b,z}]\mathbf{1} + \boldsymbol{\lambda}_{\beta^{a_b,z}})}_{\boldsymbol{\lambda}'_{\beta^{a_b,z}}} |P_{a_b,z} b - P_{a_b,z} \hat{b}|. \tag{3}$$

Then, by taking any point $\hat{b}$ and denoting $\hat{\beta}^{a,z}$ the point minimising $U(\hat{b}^{a,z})$ for each pair $(a, z)$, we have

$$U_{\hat{\beta}^{a,z}}(\hat{b}^{a,z}) \leq U_{\beta^{a,z}}(\hat{b}^{a,z}). \tag{4}$$

Using the previous intermediate results and first applying Bellman's optimality operator at $\hat{b}$, we then get:

$$V^*(\hat{b}) \leq K_{\hat{b}}^U(\hat{b})$$

$$= \max_a \left( \rho(\hat{b}, a) + \gamma \sum_z \|P_{a,z} \hat{b}\|_1 U_{\hat{\beta}^{a,z}}(\hat{b}^{a,z}) \right) \qquad \text{(by definition of update operator } K_{\hat{b}}^U)$$

$$\leq \max_a \left( \rho(\hat{b}, a) + \gamma \sum_z \|P_{a,z} \hat{b}\|_1 U_{\beta^{a,z}}(\hat{b}^{a,z}) \right) \qquad \text{(by inequation 4)}$$

$$\leq \max_a \left( \rho(\hat{b}, a) + \gamma \sum_z \left( \|P_{a,z} \hat{b}\|_1 U_{\beta^{a,z}}(b^{a,z}) + \boldsymbol{\lambda}'_{\beta^{a,z}} |P_{a,z} \hat{b} - P_{a,z} b| \right) \right) \text{(by inequation 3)}$$

$$\leq \max_a \left( \rho(\hat{b}, a) + \gamma \sum_z \left( \|P_{a,z} \hat{b}\|_1 U_{\beta^{a,z}}(b^{a,z}) + \boldsymbol{\lambda}'_{\beta^{a,z}} P_{a,z} |\hat{b} - b| \right) \right) \qquad \text{(with Lemma 3)}$$

$$\leq \max_a \left( \rho(b, a) + \boldsymbol{\lambda}_r(b, a)|b - \hat{b}| + \gamma \sum_z \|P_{a,z} \hat{b}\|_1 U_{\beta^{a,z}}(b^{a,z}) + \gamma \underbrace{\left( \sum_z \boldsymbol{\lambda}'_{\beta^{a,z}} P_{a,z} \right)}_{\boldsymbol{\lambda}'_{\beta^{a,\cdot}}} |\hat{b} - b| \right)$$

$$\text{(by local Lipschitz-continuity of } \rho)$$

$$\leq \max_a \left( \rho(b, a) + \gamma \sum_z \|P_{a,z} \hat{b}\|_1 U_{\beta^{a,z}}(b^{a,z}) \right) + \max_{a'} \left( \left( \boldsymbol{\lambda}_r(b, a') + \gamma \boldsymbol{\lambda}'_{\beta^{a',\cdot}} \right) |\hat{b} - b| \right)$$

$$\leq \max_a \left( \rho(b,a) + \gamma \sum_z \|P_{a,z}\hat{b}\|_1 U_{\beta^{a,z}}(b^{a,z}) \right) + \left( \vec{\max}_{a'} \left( \boldsymbol{\lambda}_r(b,a') + \gamma \boldsymbol{\lambda}'_{\beta^{a'},\cdot} \right) \right) |\hat{b} - b|$$

(where the second maximization is component-by-component)

$$\leq K_b^U(b) + \boldsymbol{\lambda}_b^U |\hat{b} - b| \qquad \qquad \text{(where } \boldsymbol{\lambda}_b^U = \vec{\max}_{a'} \left( \boldsymbol{\lambda}_r(b,a') + \gamma \boldsymbol{\lambda}'_{\beta^{a'},\cdot} \right).$$

Thus, the cone defined by $\langle b, K_b^U(b), \boldsymbol{\lambda}_b^U \rangle$ can be added to the upper-bounding approximator $U$ in what will be the approximator $K_b^U$ ($U$ improved by a local update at $b$). $\qquad \square$

The result and proof for the lower bound are similar, even though the proof is not exactly symmetric due to the max in Bellman's optimality operator.

Figure 3: 4x3.95

Figure 4: 4x4.95

# B Graphs

The present section contains graphs presenting, one benchmark problem at a time, the evolution of $L(b_0)$ and $U(b_0)$ as a function of time (in seconds) for each applied algorithm.

Figure 5: cheese.95

Figure 6: cit

Figure 7: fourth

Figure 8: hallway

Figure 9: hallway2

Figure 10: milos-aaai97

Figure 11: mini-hall2

Figure 12: mit

Figure 13: network

Figure 14: paint.95

Figure 15: pentagon

Figure 16: shuttle.95

Figure 17: sunysb

Figure 18: tiger70

Figure 19: tiger-grid

Figure 20: grid-info k$x$

Figure 21: grid-info k$y$

Figure 22: grid-info u$x$

Figure 23: grid-info u$y$