[Reviews · NeurIPS 2018]

Reviewer 1



This paper studies rho-POMDP, where the reward function maps a pair of a belief state and an action to a real value. The authors show that the value function is Lipschitz continuous (locally with vector Lipschitz constants) if the reward function is Lipschitz continuous. This property is then exploited to design a planning algorithm, similar to Heuristic Search Value Iteration (HSVI), where an upper bound and a lower bound are iteratively updated with the representation using sets of cones. Numerical experiments show that the algorithm works as expected. Although I have not checked the proofs, the theorems make intuitive sense and lead to an HSVI-like algorithm, which is shown to work as expected in numerical experiments. The HSVI-like algorithm exploiting Lipschitz continuity is a first of a kind, and this novelty is the strength of this paper. Although the practical merit of the proposed algorithm itself is limited, improvements along the lines of point-based approaches studied for POMDP can be expected as future work. Unfortunately, writing is rather sloppy, and some critical points are unclear in the paper. For example, I have no idea what is stated in Line 234-236. The incremental variant from Line 208 is also unclear. As far as I understood, this incremental variant is a heuristic to reduce computational complexity at the expense of not being exact. Experiments are not well designed. The focus of the experiments appears to be the comparison of different heuristics of incremental variant, but I do not think this is the incremental variant is the main topic of this paper. It is difficult to see what I should take away from these experiments. It appears that the proposed algorithm runs as expected, which however can be shown much more clearly with simpler experiments.

Reviewer 2



The paper addresses the problem of rho-POMDPs non-convex reward functions, proving that indeed under some cases they, and their resulting value functions, are Lipschitz-continuous (LC) for finite horizons. The paper also proposes and uses a more general vector form of LC, too. This result allows value function approximations of the optimal V^* to be used, as well as upper and lower bounds (U and L) on value as in HSVI, and a wide array of new algorithms to be developed. This is analogous to the PWLC result for standard POMDPs, as LC is more general, allowing for similar contraction operators with Banach's fixed point theorem as in (PO)MDPs, and finite horizon approximations of the infinite horizon objective criteria. Once the paper establishes the main result, it discusses approximations of U and L using min or max, respectively, over sets of cones. This enabled a form of HSVI for rho-POMDPs. While the contribution is primarily theoretical, the experiments compare classical PWLC/PW/LC HSVI with a variety of incremental HSVIs constrained to Lipschitz constant lambda. These incremental refinements of lambda, often starting at 1, differ in their termination criteria. They are compared on several small benchmark domains for standard POMDPs, as well as a rho-POMDP called grid-info. While the PWLC HSVI performs quite well, the incremental LC versions perform similarly, but of course also work on rho-POMDP problems. The paper does a good job of describing the rho-POMDP and the issues regarding the importance of its reward's and value's Lipschitz continuity. While the experiments do not add too much, they do confirm that the new HSVI algorithm for rho-POMDPs, backed by theory, works as expected. The main contribution is theoretical in nature and useful for rho-POMDPs. The supplement's proofs are clear and seem correct after being read in moderate detail. A minor question about the use outside a rho-POMDP: 1. What are the primary hurdles that prevent this result for general continuous state MDPs, assuming compact S and A? Perhaps similar Lemmas are needed (e.g., Lemma 3)? Just curious to what extent this and other extensions have been considered. Overall, the contribution is useful for the research on rho-POMDPs and perhaps beyond, maybe to continuous MDPs and some multi-agent MDPs (as suggested in the paper). The PWLC used in point-based methods seems likely to remain the dominant perspective on approximating standard POMDPs, supported by the experiments in the paper. The paper was clear and easy to read, with the proper mathematical rigor necessary to produce the result. Some minor comments: - Line 101: "...is greather than..." -> "...is greater than...". - Line 108: "...propose an other generalization..." -> "...propose another generalization...". - Line 114: "...(but bounds are loser)." -> "...(but bounds are looser).". - Line 178: "...cones. Wlog, we consider..." -> "...cones. We consider...". - Line 265: "...Lipschitz-continuity is here not..." -> "...Lipschitz-continuity here is not...". - Line 321: "...(IJCAI)..." -> "IJCAI-03". (Check others. This is to optionally match other references like Line 291.) - Line 354: "$f : \mathbb{R}^+^n \mapsto \mathbb{R}$" -> "$f : \mathbb{R}^+^n \rightarrow \mathbb{R}$"? - Line 358: "$P_{a,z}$ being positive..." -> "$P_{a,z}$ being non-negative..."?

Reviewer 3



post-rebuttal: I have a small suggestion for the authors to include the related work in the paper if they can and to put a small experiment in the paper where surveillance is the goal of the agent. This is the main motivation of the paper. Overall I am happy with the paper and response. ----- Summary: This paper presents a method for solving POMDPs (offline POMDP planning) when the value function of the POMDP is not necessarily convex (mainly breaking the PWLC property exploited by most POMDP planners). The paper presents a new POMDP planner based on heuristic search value iteration that does not rely on convexity of the optimal value function to compute it. The idea is to use upper and lower bound on the value function (and then tighten them at various beliefs) to compute the optimal value function (or an approximation of it). The property that the paper exploits to generate these upper and lower bounds is that the optimal value function has to be Lipschitz continuous. The paper also shows that if the immediate expected reward is Lipschitz continuous then the optimal value function is guaranteed to be Lipschitz continuous and then exploits this property of the value function to propose upper and lower bounds on the value function by obtaining an expression for the Lipschitz contant for the curves (cones in this case) to upper and lower bound the value function. Finally, the paper gives empirical results on standard POMDP problems in literature. Quality - This is a good quality paper; thorough and rigorous for most parts. Clarity: The paper is clear and well-written. Originality: The paper presents original work. Significance: I think the paper presents significant results that are relevant to this community. Strength: I believe the main strength of the paper is the carefully designed and principle method for obtaining Lipschitz continuous upper and lower bound on the value function of a POMDP. Weakness: - I am quite not convinced by the experimental results of this paper. The paper sets to solve POMDP problem with non-convex value function. To motivate the case for their solution the examples of POMDP problem with non-convex value functions used are: (a) surveillance in museums with thresholded rewards; (b) privacy preserving data collection. So then the first question is when the case we are trying to solve are above two, why is there not a single experiment on such a setting, not even a simulated one? This basically makes the experiments section not quite useful. - How does the reader know that the reward definitions of rho for this tasks necessitates a non-convex reward function. Surveillance and data collection has been studied in POMDP context by many papers. Fortunately/unfortunately, many of these papers show that the increase in the reward due to a rho based PWLC reward in comparison to a corresponding PWLC state-based reward (R(s,a)) is not that big. (Papers from Mykel Kochenderfer, Matthijs Spaan, Shimon Whiteson are some I can remember from top of my head.) The related work section while missing from the paper, if existed, should cover papers from these groups, some on exactly the same topic (surveillance and data collection). - This basically means that we have devised a new method for solving non-convex value function POMDPs, but do we really need to do all that work? The current version of the paper does not answer this question to me. Also, follow up question would be exactly what situation do I want to use the methodology proposed by this paper vs the existing methods. In terms of critisim of significance, the above points can be summarized as why should I care about this method when I do not see the results on problem the method is supposedly designed for.